

# A high-resolution map of diffuse groundwater recharge rates for Australia

Stephen Lee[1,2], Dylan J. Irvine[1,2], Clément Duvert[1,2], Gabriel C. Rau[3,2], Ian Cartwright [4,2]

[1]Research Institute for the Environment and Livelihoods, Charles Darwin University, Darwin, 8000, Australia
[2]National Centre for Groundwater Research and Training, Adelaide, 5000, Australia
[3]School of Environmental and Life Sciences, The University of Newcastle, Callaghan, 2308, Australia
[4]School of Earth, Atmosphere and Environment, Monash University, Clayton, 3800, Australia

*Correspondence to*: Stephen Lee (stephen.lee@cdu.edu.au)

**Abstract.** Estimating groundwater recharge rates is important to understand and manage groundwater. Numerous studies have used collated recharge datasets to understand and project regional or global-scale recharge rates. Recharge estimation methods each have distinct assumptions, quantify different recharge components, and operate over different temporal scales. To address these challenges, we use over 200,000 groundwater chloride measurements to estimate groundwater recharge rates using the chloride mass balance (CMB) method across Australia. Recharge rates were produced stochastically using gridded chloride
deposition, runoff, and precipitation datasets. After filtering out recharge rates where the assumptions of the method may have been compromised, 98,568 estimates of recharge were produced. The resulting recharge rates and 17 spatial datasets were integrated into a random forest regression algorithm, generating a high-resolution (0.05°) model of recharge rates across Australia. The regression reveals that climate-related variables, including precipitation, rainfall seasonality, and potential evapotranspiration, exert the most significant influence on recharge rates, with vegetation (NDVI) also contributing
significantly. Importantly, both the mean values of the recharge point dataset (43.5 mm y$^{-1}$) and of the spatial recharge model (22.7 mm y$^{-1}$) are notably lower than those reported in previous studies, underscoring the prolonged timescale of the CMB method and the potential disparities arising from distinct recharge estimation methodologies. This study presents a robust and automated approach to estimate recharge using the CMB method, offering a unified model based on a single estimation method. The resulting datasets, the Python script for recharge rate calculation, and the spatial recharge models collectively provide
valuable insights for water resources management across the Australian continent and similar approaches can be applied globally.



## 1 Introduction

Groundwater is a critical component of the water cycle, providing baseflow to streams and supporting ecosystems and livelihoods (Brunke and Gonser, 1997; Eamus, 2006; Shah, 2005). With impacts from climate change, population growth and increased usage, groundwater resources are expected to become even more important in the future (Döll, 2009; Famiglietti, 2014; Wada et al., 2010), requiring a detailed understanding of hydrogeological processes through desktop studies, numerical modelling, and direct field measurements. Assessing groundwater resources not only requires understanding their distribution,

natural discharge and extraction rates, but also mechanisms and rates of resource replenishment.

Groundwater recharge is one of the most important, albeit challenging, components to quantify in groundwater assessments due to its wide spatiotemporal variability, which is influenced by a range of geo-eco-climatic factors (de Vries and Simmers, 2002). Recharge estimation is further complicated by the conceptualisation of recharge mechanisms (e.g., diffuse versus focused; Lerner et al., 1990). Similarly, the uncertainties of recharge estimation techniques provide further challenges (Scanlon

et al., 2002). Additional complexities need to be carefully considered in recharge studies, including understanding the timescales associated with the technique(s) being used (e.g., Scanlon et al., 2002; Cartwright et al., 2017) and the component of recharge being estimated (e.g., gross, potential, or net recharge; Crosbie et al., 2010a).

Large scale studies of groundwater recharge (e.g., global and continental scale) that are based on the compilation of recharge estimates, typically utilise recharge estimates obtained from different techniques (e.g., Petheram et al., 2002; Scanlon et al.,

2006; Crosbie et al., 2010a; Mohan et al., 2018; Moeck et al., 2020; MacDonald et al., 2021; Berghuijs et al., 2022). These combined datasets allow an assessment of the changes in recharge rates over time due to climate variability or land cover change (e.g., Scanlon et al., 2006). However, such datasets add extra uncertainty to the predictive models that utilise them, given that they include recharge estimates with different assumptions, temporal scales, and mechanisms (e.g., Crosbie et al., 2010a; Mohan et al., 2018). Utilising different recharge estimation techniques may result in widely different recharge rates

(e.g., Crosbie et al., 2010a; King et al., 2017; Walker et al., 2019; Cartwright et al., 2020).

Selecting recharge estimates from a single technique from these global studies could overcome the issues mentioned above, but also lead to insufficient spatial coverage for meaningful continental-scale assessments. For example, the issue of spatial coverage of recharge estimates is evident in Australia from the sparseness of recharge estimates in the interior of Australia (e.g., Moeck et al., 2020; Berghuijs et al., 2022). Studies in Australia have addressed the issue of data sparsity through creation

of a series of empirical relationships between rainfall and recharge by investigating key factors such as vegetation and soil types (e.g., Crosbie et al., 2010a; Leaney et al., 2011). More recent Australian studies have utilised statistical methods to investigate the influence of environmental variables on groundwater recharge (e.g., Fu et al., 2019) or applied machine learning techniques to predict future recharge (e.g., Huang et al., 2019, 2023). Others have focused on upscaling of point estimates from a single technique (e.g., chloride mass balance) to a regular grid across regional study areas using regression kriging (e.g.,

Crosbie et al., 2018; Crosbie and Rachakonda, 2021; Crosbie et al., 2022).



The chloride mass balance (CMB) method is one method that provides the opportunity for detailed studies of diffuse groundwater recharge rates, given the wide availability of groundwater chloride concentration measurements. The CMB method is also the most widely used recharge estimation technique globally (Moeck et al., 2020), in semi-arid and arid regions (Scanlon et al., 2006), and in Australia (e.g., Crosbie and Rachakonda, 2021; Crosbie et al., 2018, 2010a, b; Petheram et al.,
2002). The CMB method provides long-term estimates of diffuse recharge over the timescale required for chloride to accumulate in the subsurface, which ranges from years to decades in temperate settings (Cartwright et al., 2020), and up to thousands of years in semi-arid and arid areas (Scanlon et al., 2002, 2006). Spatially, the CMB method estimates diffuse recharge over the areas upgradient from the measurement location, ranging from a few hundred metres to several kilometres (Scanlon et al., 2002). The generation of chloride deposition maps (e.g., Davies and Crosbie, 2018; Wilkins et al., 2022) have
allowed for the large-scale (regional) use of the CMB method (e.g., Crosbie et al., 2018). Irvine and Cartwright (2022) utilised the chloride deposition maps from Davies and Crosbie (2018) to automate the application of the CMB method in Python. Automating the application of the CMB method provides opportunities for large datasets of recharge to be efficiently generated from chloride measurements.

This study utilises recently developed chloride deposition maps from Wilkins et al. (2022) and approaches to automate analyses
to estimate long-term diffuse groundwater recharge rates based on the CMB method across the Australian continent. We collate a large dataset of groundwater chloride and associated spatial datasets to facilitate the recharge estimates. We utilise these datasets and the random forest algorithm to develop a regression model for long-term diffuse groundwater recharge rate estimation for the Australian continent. Using the model, we explore the control of environmental variables on groundwater recharge rates, quantify the uncertainty in recharge rate predictions and produce point datasets and high-resolution gridded
maps of diffuse recharge for Australia.

## 2 Methods

### 2.1 Collation of groundwater chloride dataset

Groundwater chloride measurements were collated from the following sources: the Geoscience Australia Portal (Geoscience Australia, 2022); the CSIRO Hydrogeochemical Mapping of the Australian Continent series dataset (Gray et al., 2019; Gray
and Bardwell, 2016a, b, c, d, e, f; Henne and Reid, 2021); a dataset collated for the state of South Australia (Broad, 2020); Visualising Victoria's Groundwater (FedUni, 2022); and a Northern Territory Government isotope dataset (Tickell, pers. Comm., 12 April 2022). The preliminary collated dataset contained a total of 226,954 chloride measurements (including bores with time series data and duplicate values). A breakdown of the individual counts of each dataset compiled is provided in Table S1 of the supporting information.
Bore log information was downloaded from the Australian Groundwater Explorer (Bureau of Meteorology, 2022c) to provide location, bore hole depths, drilled depths, and screened interval depths. The depth assigned for each chloride measurement was





applied in the following order of preference: screen mid-point depth, sample depth, bore depth, and hole depth. Measurements with no depth information were removed from the analyses.

Several preliminary measures were undertaken for quality assurance of the chloride data. All measurements without a latitude
and longitude were removed. Chloride measurements that were reported below the analytical detection limit (i.e., <1 mg L$^{-1}$) were removed from the dataset. All duplicates with matching bore identifiers, latitude, and longitude (in decimal degrees), sample date, and chloride concentration were presented as a single measurement, resulting in 192,300 measurements. Measurements without a sample date were retained because excluding them would remove 99.8 % of measurements from the state of Western Australia ($n = 19,967$).

Bores with repeat measurements from different sample dates were represented as the mean of the time series, producing a final dataset with 115,630 bores each with a single chloride value for the analyses. Due to the size of the dataset, analysis of charge balance errors was not undertaken in this study. The final chloride dataset is provided as a downloadable electronic data file in the supporting information.

**2.2 Collation of spatial datasets**

To investigate factors that influence groundwater recharge, we identified 17 different gridded datasets (Table 1). These variables were chosen based on their use in previous global groundwater recharge studies (e.g., Mohan et al., 2018; Moeck et al., 2020) or in regional scale to continental-scale recharge studies in Australia (e.g., Crosbie et al., 2010a; Leaney et al., 2011). All analyses in our study utilise the native resolution of the datasets shown in Table 1.

**Table 1.** Spatial datasets of factors that are known to influence groundwater recharge. Variables are grouped into climatological-related, surface process and hydrogeological-related, geomorphological-related, and vegetation-related datasets. AHD denotes the Australian Height Datum.

| Variable (symbol) | Unit | Resolution | Description | Reference |
|---|---|---|---|---|
| **Climatological** | | | | |
| Precipitation (P) | mm y$^{-1}$ | 0.05° × 0.05° | The mean annual P, PET and aridity index were calculated by averaging data from 21 overlapping decadal periods spanning from 1911 to 2020. | Bureau of Meteorology (Bureau of Meteorology, 2023b) |
| Potential evapotranspiration (PET) | mm y$^{-1}$ | 0.05° × 0.05° | | Bureau of Meteorology (Bureau of Meteorology, 2022d) |
| Aridity index (P/PET) | - | 0.05° × 0.05° | | Bureau of Meteorology (Bureau of Meteorology, 2023b; |



| | | | | Bureau of Meteorology, 2022d) |
|---|---|---|---|---|
| Köppen Geiger classification | - | 0.0833° × 0.0833° | Climate classification for the present-day, from 1980 to 2016. | Beck et al. (2018) |
| Rainfall seasonality (all zones) | - | 0.25° × 0.25° | Based on median annual rainfall and seasonal incidence from 1900 to 1999. | Bureau of Meteorology (Bureau of Meteorology, 2022a) |
| **Surface processes and hydrogeological** | | | | |
| Ground elevation | m AHD | 0.0008° × 0.0008° | Geoscience Australia SRTM 3 sec DEM version 1. | Gallant et al. (2009) |
| Depth to water table | m | 0.008° × 0.008° | Output of global numerical groundwater model. | Fan et al. (2013) |
| Regolith depth | m | 0.0008° × 0.0008° | Soil and landscape grid national soil attribute maps – depth of regolith (3 arc sec resolution) version 6. | Wilford et al. (2018) |
| Slope | % | 0.0008° × 0.0008° | CSIRO data published in 2016. Slope derived from 1 sec SRTM DEM-S version 4. | Gallant and Austin (2012) |
| Distance to coast | km | - | Not a national gridded dataset. Calculated using GEODATA Coast 100K 2004 coastline and the Distance Matrix tool in QGIS. | Geoscience Australia (2004) |
| **Geomorphological** | | | | |
| Sand fraction | % | 0.0008° × 0.0008° | CSIRO data published in 2022 as release 1 version 6 (sand and silt) and release 2 version 4 (clay). 100 to 200 cm interval. | Malone and Searle (2022b) |
| Silt fraction | % | 0.0008° × 0.0008° | | Malone and Searle (2022c) |
| Clay fraction | % | 0.0008° × 0.0008° | | Malone and Searle (2022a) |



| Geology | - | 0.001° × 0.001° | Surface Geology of Australia 1:1M scale categorised into simpler groups. | Raymond et al. (2012) |
|---|---|---|---|---|
| Australian Soil Classification | - | 0.0025° × 0.0025° | Australian Soil Resource Information System Australian Soil Classification. | CSIRO (CSIRO, 2023) |
| **Vegetation related** | | | | |
| NDVI | - | 0.05° × 0.05° | Indicator of vegetation greenness. Values presented as the mean of the 3-monthly averages from July 1992 to January 2019. | Bureau of Meteorology (Bureau of Meteorology, 2022e) |
| Vegetation class (major) | - | 0.0009° × 0.0009° | Present (extant) major vegetation groups from the National Vegetation Information System. Categorised based on Eamus et al. (2016). | Department of Climate Change, Energy, the Environment and Water (Department of Climate Change, Energy, the Environment and Water, 2022) |

The decadal rainfall maps from the Bureau of Meteorology (2023b) were chosen over the Australian Water Outlook

precipitation data (Bureau of Meteorology, 2022d) used in the Australian Water Resources Assessment Landscape (AWRA-L) model (Frost and Shokri, 2021), due to missing and unreliable data in the Australian Water Outlook dataset for a large area of north-central Western Australia and other smaller areas in South Australia and Northern Territory. Non-gridded spatial data were also used, including the Australian coastline (Geoscience Australia, 2004; for the purposes of approximating the distance from bore holes to the coast; Table 1) and a halite deposit dataset of Australia (Feitz et al., 2019).

Spatial maps of the variables from Table 1 and the halite deposit are provided as Figure S1 in the supporting information.

To assist with later assessments, all gridded spatial data collated in Sect. 2.2 (Table 1) were appended to the recharge output produced later in Sect. 2.3. The Point Sampling Tool in QGIS was used to extract the corresponding value from the raster pixel in which the groundwater recharge rate derived from CMB is located. The Distance Matrix tool in QGIS was used to measure the nearest distance to the Australian coastline. Some groundwater recharge rates were located outside of the extents of some

gridded spatial data.

To produce a continental scale recharge estimator, all spatial resolutions were converted to a 0.05-degree grid. For conversion, the GDAL Warp (reproject) tool in QGIS was used, utilising the average resampling method. The average resampling method



was chosen as opposed to one of the more commonly used methods that take the value or aggregation of a limited number of nearest pixels (e.g., nearest neighbour, bilinear interpolation or cubic convolution). The average method considers all pixels

that contribute to the output pixel in its calculation, preserving the overall statistical characteristics of the data, while producing a smooth output (similar to cubic convolution), and covering areas of the coastline that were not observed when using other resampling methods.

### 2.3 Chloride Mass Balance analysis

The CMB method produces estimates of long-term groundwater recharge by comparing groundwater (or soil water) chloride

concentration to that measured in rainfall (and dry deposition), provided various assumptions are met (Wood, 1999; Leaney et al., 2011). The method assumes that chloride acts conservatively, is solely sourced from precipitation, and that groundwater has returned to steady-state conditions following any land-use changes (e.g., vegetation clearing; Leaney et al., 2011). Following Davies and Crosbie (2018), recharge (R, mm y$^{-1}$) from the CMB method can be calculated using Eq. 1:

$$R = \frac{100D}{\text{Cl}_{\text{gw}}},\qquad\qquad\qquad(1)$$

where D is the chloride deposition rate due to rainfall (kg ha$^{-1}$ y$^{-1}$), Cl$_{\text{gw}}$ is the chloride concentration in groundwater (mg L$^{-1}$), and a multiplier of 100 is applied for unit conversion.

While Eq. 1. assumes that no chloride is exported laterally, the input/output of chloride through runoff or run-on can be accounted for by modifying Eq. 1 (e.g., Crosbie et al., 2018). Accounting for lateral export of chloride can be especially important in upland areas with steep topography and high rainfall (Leaney et al., 2011). The uncertainty associated with run-

on is suggested to be negligible (e.g., Crosbie et al., 2018), while the uncertainty associated with chloride concentration in runoff is small compared to that of chloride deposition (Leaney et al., 2011). However, due to the large number of bores, and the continental scale of this study where a range of landscapes may be covered, runoff was accounted for to address this uncertainty. Following Crosbie et al. (2018) and Crosbie and Rachakonda (2021), the modified Eq. 2 can be used:

$$R = \frac{100\,D(1-\alpha\cdot RC)}{\text{Cl}_{\text{gw}}},\qquad\qquad\qquad(2)$$

where RC (-) is the runoff coefficient determined by dividing the long-term average annual runoff by the long-term average annual precipitation, and α is a scalar.

In this study, we used a modified version of the Chloride Mass Balance Estimator for Australian Recharge (CMBEAR; Irvine and Cartwright, 2022). The modified version of CMBEAR utilises the Australian gridded dataset of chloride deposition (i.e., Wilkins et al., 2022) to automate recharge estimation using the CMB method. The modified version also applies Eq. 2, where

the previous version applied Eq. 1. In this updated version of CMBEAR, when applying Eq. 2 uncertainty for each input variable is quantified using a stochastic approach adopted from Crosbie et al. (2018).



Out of 115,630 bores in our dataset, 79 % only had one groundwater chloride measurement available. To estimate an uncertainty in groundwater chloride, bores with more than 10 measurements ($n$ = 1,516) were used to calculate a mean coefficient of variation (CVμ). As per Crosbie et al. (2018), the coefficient of variation was calculated for each bore, with the

resulting CVμ being the mean of these values. The CVμ of 0.37 was multiplied by the mean chloride value ($Cl_{gw}μ$) for each bore in our dataset to estimate the standard deviation ($Cl_{gw}σ$). The $Cl_{gw}μ$ and $Cl_{gw}σ$ were then used to generate normal distributions for each bore. A normal distribution was adopted because 52 % of bores with more than 10 measurements passed a normality test (p-value >0.05). The approach to use the CV, rather than a standard deviation directly was made since the CV scales with the mean chloride value, whereas applying the same standard deviation to all values could be problematic for small

values (i.e., values becoming negative).

For each bore, the mean, standard deviation, and skew of the chloride deposition (Dμ, Dσ and Dskew, respectively) were extracted from the chloride deposition map from Wilkins et al. (2022) from the pixel in which the bore was located and used to generate a Pearson Type III distribution, following the description from Wilkins et al. (2022).

While the RC extracted from the location of the bore is held constant, this value is scaled down by the α value (Eq. 2) which

is sampled from a uniform distribution between 0.33 and 0.66. This scaling approach is adopted from Crosbie et al. (2018) to deal with uncertainty in the proportion of baseflow contributing to runoff, and the below average chloride concentration in high intensity rainfall events that typically generates runoff. Long-term annual runoff was calculated by averaging annual runoff data from 21 overlapping decadal periods spanning from 1911 to 2020 (Bureau of Meteorology, 2023b). As this runoff data was an output from the AWRA-L model (Frost and Shokri, 2021) and reliant on precipitation inputs which contained

missing and unreliable values (see Sect. 2.2), the runoff data was therefore unreliable in certain areas. The problematic areas were identified as those with long-term annual precipitation <100 mm y$^{-1}$ (Bureau of Meteorology, 2022d). A mask for the RC dataset was created using these areas and used to convert all RC values in problematic areas to 0.0018 (the minimum RC calculated for an adjacent rectangular area covering similar latitudes compared to the problematic areas, from -29.5 to -20.5 degrees, and longitudes from 133.0 to 136.0 degrees). Long-term average annual precipitation was calculated from decadal

rainfall maps (Bureau of Meteorology, 2023b) as mentioned in Table 1. While further investigation into the range and distribution type for the α value could be conducted, the range used has been used across multiple climate zones (e.g., Crosbie et al., 2018; Crosbie and Rachakonda, 2021; Crosbie et al., 2022).

A probability distribution was created for each bore by calculating recharge (R) 1,000 times using the 1,000 sampled replicates from the distributions of $Cl_{gw}$, D and α. The median recharge ($R_{50}$), 95$^{th}$ percentile recharge ($R_{95}$) and 5$^{th}$ percentile recharge

($R_5$) values were calculated from each probability distribution and provided as outputs for each bore. The median was chosen as it is unaffected by extreme outliers as is with the arithmetic mean.





## 2.4 Data filtering

The assessment of the suitability of input data for the application of the CMB method is a vital step to ensure that the assumptions of the method are met (Irvine and Cartwright, 2022). In our study, this assessment (hereafter referred to as data
filtering process) involved six steps that were performed after obtaining the recharge estimates.

The data filtering process removed recharge estimates where the following conditions likely invalidate the CMB method, or where unrealistic recharge estimates were produced using the following steps:

(1) bores where the screen mid-point is ≥150 m below ground surface (bgs) which are unlikely to be in an unconfined aquifer (e.g., Crosbie and Rachakonda, 2021; Crosbie et al., 2022);

(2) bores with mean chloride concentrations <2 mg L$^{-1}$ are unlikely to be representative of groundwater where poor bore construction allows rain water to rapidly reach the well screen (e.g., Crosbie and Rachakonda, 2021; Crosbie et al., 2022);

(3) bores with both mean chloride concentration ≥2,000 mg L$^{-1}$ and where depth to the water table ≤1 m bgs are likely to be in or downstream of discharge areas (criteria modified from Crosbie and Rachakonda (2021) and Crosbie et al.
200   (2022));

(4) bores located within the known area of the Amadeus Basin halite deposit which could be a potential additional source of chloride;

(5) bores that are located <1 km from the coast may contain additional chloride from marine sources, and are in coastal areas prone to large chloride deposition variability and uncertainty;

(6) cases where estimated recharge equals or exceeds mean annual rainfall were also removed (e.g., West et al., 2023).

The outcomes of the data filtering process are provided both in Sect. 3.2 and in more detail in the supporting information.

## 2.5 Random forest analyses

Random forest analyses have been utilised for a wide range of applications in hydrogeological studies, including predictive modelling of groundwater pollutants (e.g., Rodriguez-Galiano et al., 2014; Ouedraogo et al., 2019), source aquifer attribution
of hydrogeochemical samples (e.g., Baudron et al., 2013), modelling groundwater levels (e.g., Koch et al., 2019), modelling groundwater potential (e.g., Rahmati et al., 2016), and predicting groundwater recharge (e.g., Sihag et al., 2020; West et al., 2023). In this study, we implemented the random forest regressor from the Scikit-learn Python library (Pedregosa et al., 2011) to develop groundwater recharge prediction models.

Our dataset comprised groundwater recharge as the target variable and 17 influential factors (i.e., spatial variables from Table
1). These factors were utilised for feature importance analyses and to produce a model to predict recharge. The random forest feature importance provides insight into how each input variable contributes to the predictive performance of the random forest model. The feature importance for a variable is generated according to the mean decrease in variance produced by including that variable at a split in the decision tree.



Three models were produced, using $R_{50}$, $R_{95}$, and $R_5$ long-term annual recharge produced from the CMB analysis. The dataset
was split into a randomly selected training subset (70 %) and validation subset (remaining 30 %) following typical practice
(e.g., West et al., 2023; Sihag et al., 2020; Rahmati et al., 2016). Each tree in the random forest model (the model) was trained
on $n$ randomly selected observations, with replacement (i.e., bootstrapping) from the training subset, where $n$ is equal to the
total number of observations in the training subset. The observations chosen to train the model are referred to as 'in-the-bag'
samples whereas those not chosen are known as 'out-of-bag' samples (Cutler et al., 2012). The random forest algorithm
introduces further randomness at each split in a tree by random selection of a subset of the total number of input variables
(Pedregosa et al., 2011). Once a model was trained, external validation was conducted by making predictions using the reserved
validation subset. The locations of bores used in the training and validation datasets are provided in Figure S3 of the supporting
information.

Multiple models were produced using $R_{50}$ as the target variable, and various combinations of the 17 input features to determine
the impact of the choice of input features on model performance. The grid search with cross validation method was used to
determine the best values to use for hyperparameters including maximum depth, maximum features, minimum samples in a
leaf, and minimum samples per split (Pedregosa et al., 2011). No limit was set for maximum leaf nodes as per the default
random forest regressor settings from the Scikit-learn Python library (Pedregosa et al., 2011). Each model was run using 50,
100, 150, 200, 250, 300, 350, and 400 trees. The performance of a model was assessed through goodness-of-fit using the
training score, i.e., the Pearson $R^2$ value obtained from comparing the point recharge training data value versus modelled
recharge value.

An external validation of the model was performed by running predictions on the 30 % of data that was reserved for testing
the model. A test score ($R^2$) was obtained through comparing point versus modelled recharge. An internal validation of the
model was performed by running predictions for the 'out-of-bag' samples in trees for which those samples were not used for
training. An 'out-of-bag' prediction score ($R^2$) was obtained. The model with the highest test score was further evaluated
through its training score to assess whether the model was 'over-fitting'. Hyperparameters were adjusted accordingly to reduce
the difference between the training score and test score to limit over-fitting. The optimal number of trees to use in the model
was determined as the point when increasing the number of trees did not increase the 'out-of-bag' score. Cross-validation was
also conducted on the training subset through a k-fold test with 10 folds to ensure the model was not biased by data selection.
The feature importance tool was used to determine the relative importance of each input feature in our random forest model.
Finally, three gridded recharge maps ($R_5$, $R_{50}$ and $R_{95}$) were produced using the optimal combination of spatial variables and
trees as initially explored using $R_{50}$.





# 3 Results

## 3.1 Distribution of chloride measurements

The $Cl_{gw}$ data collated in this study and its distribution are shown in Figure 1. $Cl_{gw}$ varies widely across the Australian continent, ranging from 1 mg L$^{-1}$ to >200,000 mg L$^{-1}$ (Figure 1a). Moderate to high $Cl_{gw}$ concentrations predominantly occur in inland Australia. High $Cl_{gw}$ concentrations are particularly prominent in southern Australia, in areas including the Murray Darling Basin near the South Australia-Victoria-New South Wales junction where dryland salinity issues have been reported (e.g., Cartwright et al., 2007). Other $Cl_{gw}$ hotspots such as in southern Western Australia correspond with where salt lakes exist (e.g.,

Bowen and Benison, 2009). As expected, the lowest $Cl_{gw}$ concentrations are mainly located in the monsoon-influenced tropical north of Australia and along much of the temperate east coast of Australia where rainfall is typically high (>1,000 mm y$^{-1}$; Figure 1a).

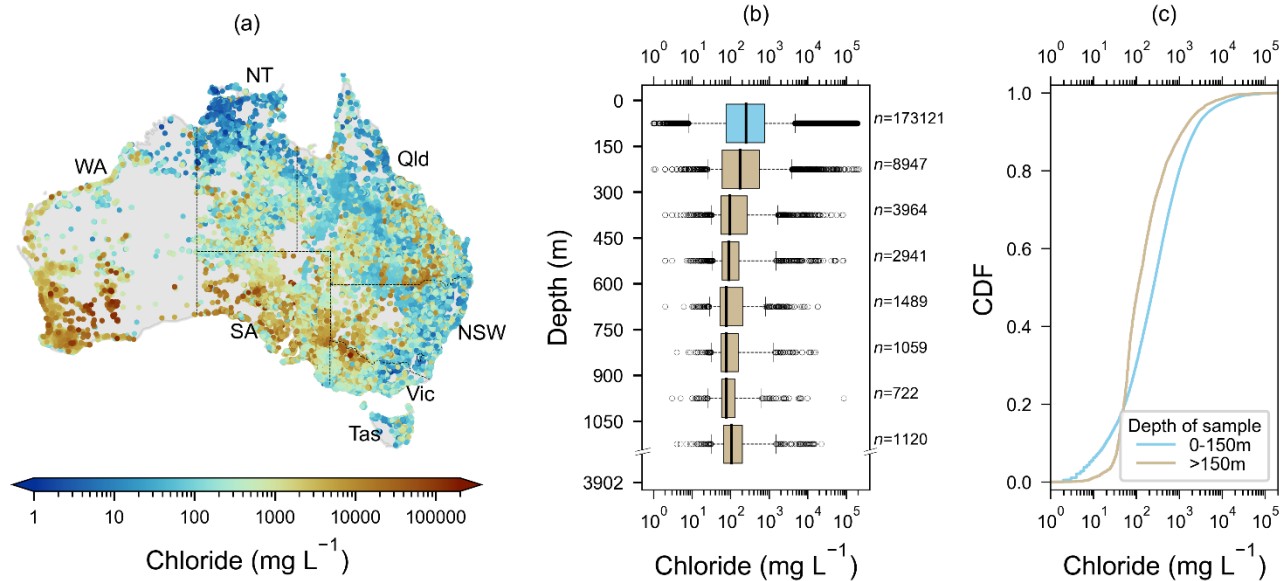

**Figure 1.** Spatial distribution of groundwater chloride ($Cl_{gw}$) shown as **(a)** locations and concentrations of $Cl_{gw}$, with Australian states and
territories marked as NT (Northern Territory), Qld (Queensland), NSW (New South Wales), Vic (Victoria), Tas (Tasmania), SA (South Australia) and WA (Western Australia); **(b)** box plots showing the depth distribution of $Cl_{gw}$. Box plots were binned by 150 m depth intervals except for the last box which contains $Cl_{gw}$ measurements sampled from a depth of >1,050 m. The blue box corresponds to the data used for recharge estimation. The upper and lower extents of the boxes represent the 75th and 25th percentiles of $Cl_{gw}$, respectively. The upper and lower whiskers represent the 95th and 5th percentiles of $Cl_{gw}$, respectively. The medians are shown as black lines and outliers are shown as
hollow black circles; **(c)** cumulative distribution function (CDF) of $Cl_{gw}$ for shallow wells (depth of sample from 0–150 m) and deep wells (>150 m).

Figure 1b shows the variation of chloride with depth. Most of the data are within 150 m of the ground surface ($n = 171,681$; median $Cl_{gw}$: 250 mg L$^{-1}$). The median $Cl_{gw}$ decreases with depth between 0 and 900 m, followed by an increase between 1,050 and 3,902 m. This notably contrasts with other regions in the world (e.g., Ferguson et al., 2023) due to Australia's unique

climatic and geologic conditions (see Figure S2 in supporting information for more details).





The cumulative distribution function (CDF) plot (Figure 1c) shows the difference in $Cl_{gw}$ distribution between shallow (<150 m) and deep (>150 m) bores in Australia, with the shallow bores spanning a much wider range of $Cl_{gw}$ values compared to deeper bores. The CDF plot also highlights the proportionally lower number of low $Cl_{gw}$ values (47 % of deep bores have $Cl_{gw}$ <100 mg $L^{-1}$) and a lower median value of deeper bores (median $Cl_{gw}$ = 110 mg $L^{-1}$) compared to shallow bores (30 % of shallow bores have $Cl_{gw}$ <100 mg $L^{-1}$; median $Cl_{gw}$ = 250 mg $L^{-1}$).

## 3.2 Recharge estimates and data filtering

Figure 2 shows the data filtering process applied to remove values that do not meet the assumptions required to apply the CMB method. It is important to note that the same bores that were excluded for $R_{50}$ during each step of the data filtering process (Figure 2) were also excluded for $R_5$ and $R_{95}$. The recharge dataset prior to data filtering is provided as an electronic data file in the supporting information.





**Figure 2.** Data filtering process showing all data **(a)** and the groundwater recharge rate (R; mm y$^{-1}$) estimates that were included at each step with statistics for R$_{50}$ (mean, standard deviation and number of measurements remaining) and box plots for R$_{50}$ binned by P at 200 mm y$^{-1}$ intervals (except the >1,600 mm y$^{-1}$ bin). The upper and lower extents of the boxes represent the 75$^{th}$ and 25$^{th}$ percentiles of R$_{50}$,



respectively. The upper and lower whiskers represent the 95th and 5th percentiles of $R_{50}$, respectively. The medians are shown as orange lines and outliers are shown as hollow black circles. The remaining number of measurements at each step is shown above the box plot. The maps on the right show the location of data, the number of measurements removed, and cumulative number of measurements removed at each step.

The boxplots in Figure 2 present the $R_{50}$ distribution binned by P in 200 mm y$^{-1}$ intervals (except the >1,600 mm y$^{-1}$ bin) at

each step after data filtering. P ranged from 109 mm y$^{-1}$ to 4,231 mm y$^{-1}$. The 600–800 mm y$^{-1}$ bin contained the greatest number of $R_{50}$ values (~33 %), followed by the 400–600 mm y$^{-1}$ bin (~21 %). Throughout the data filtering process, each bin was affected in different ways. $R_{50}$ values in the 400–600 mm y$^{-1}$ bin had the highest number of exclusions ($n = 5,460$ between Figure 2a and 2g). While the number of exclusions from the 0–200 mm y$^{-1}$ bin was low (n = 422), as a percentage this was a substantial cut of ~20 % to the recharge estimates within this P range.

A map visualising the spatial locations of data being removed is shown for each step of the data filtering process in Figure 2 (Figure 2, right column). While clear spatial trends could be inferred for data removed in step 1 where deep bores were removed from the dataset (e.g., mostly bores in the Great Artesian Basin), step 4 where known halite deposits were removed (e.g., Amadeus Basin halite deposit) and step 5 where bores near the coast were removed, without detailed analyses, no obvious factors could be identified from most of the other steps. A visual assessment shows that bores that were removed in step 3

broadly align with areas likely to contain areas of high hazard or risk of dryland salinity (National Land and Water Resources Audit, 2001).

At the end of the data filtering process (Figure 2g), ~12 % of the original dataset was removed, leaving 98,568 recharge values. Overall, the change in mean $R_{50}$ ($\mu R_{50}$) was minimal with ~2 % decrease from an initial $\mu R_{50}$ of 44.3 mm y$^{-1}$ to 43.5 mm y$^{-1}$. The largest change in $\mu R_{50}$ between steps was in the depth filtering step (i.e., sample depth >150 m bgs), with a 7 % increase

in $\mu R_{50}$ (Figure 2b). Removing sample depths more than 150 m bgs, is crucial because most of the deep bores are located within the Great Artesian Basin and similar deep confined aquifers. The recharge area of these deep systems is likely to be hundreds of kilometres away from the bore location, whereas our analyses assume recharge occurs within the 0.05° × 0.05° pixel from the chloride deposition map that contains the bore.

It is important to note that while the overall $\mu R_{50}$ did not change significantly at the end of the data filtering process, the

standard deviation of $R_{50}$ ($\sigma R_{50}$) decreased by ~ 40 %. The noticeable decrease in $\sigma R_{50}$ is the result of the exclusion of high recharge values generated from chloride concentrations <2 mg L$^{-1}$ in step 2 (Figure 2c), and recharge values with R/P >1 in step 6 (Figure 2g). While step 6 (Figure 2g) did not remove a significant number of $R_{50}$ values ($n = 118$), it is likely that many $R_{50}$ values with R/P >1 had already been removed in previous steps of the data filtering process due to other factors.

The resulting recharge estimates for $R_{50}$, $R_{95}$ and $R_5$ are shown in Figure 3a, b and c, respectively. The mean recharge for $R_{50}$,

$R_{95}$, and $R_5$ are 43.5 mm y$^{-1}$, 113.4 mm y$^{-1}$, and 25.8 mm y$^{-1}$, respectively.



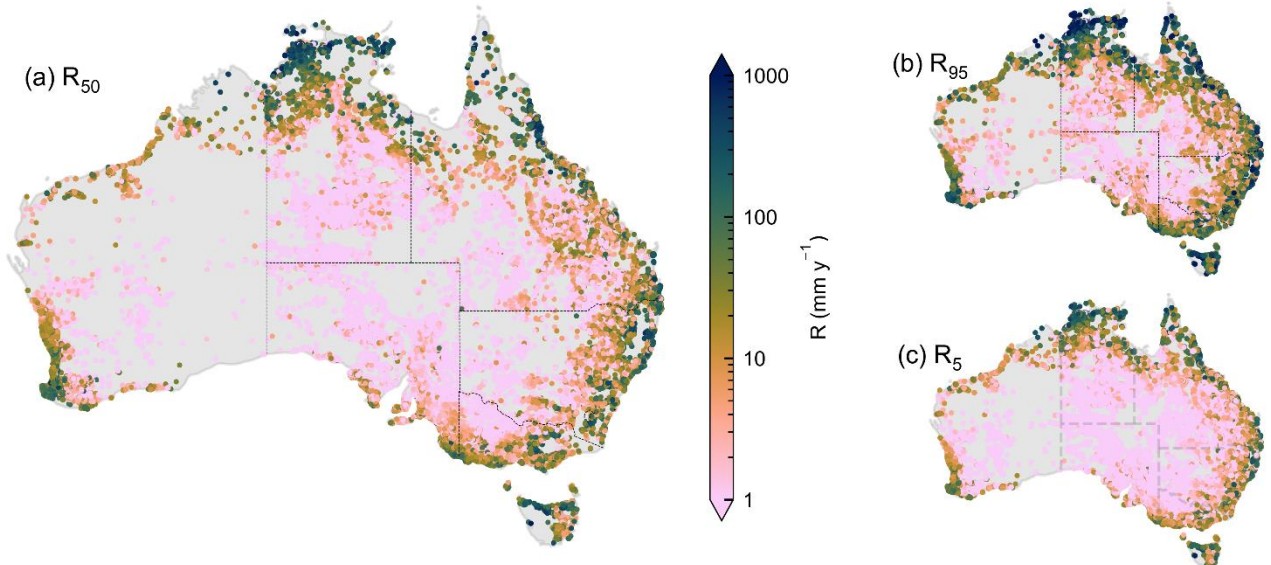

**Figure 3.** Groundwater recharge rates (R; mm y$^{-1}$) estimated using CMB from 98,568 bores. Maps show: **(a)** median recharge ($R_{50}$), **(b)** 95$^{th}$ percentile recharge ($R_{95}$) and **(c)** 5$^{th}$ percentile recharge ($R_5$) rates.

As expected, high recharge rates are mostly located in areas with high precipitation, i.e., in the tropical north, along the east

coast, and in north-western Tasmania (Figure 3), while low recharge rates are mostly located inland from the coast. However, there is variability in recharge rates, spanning 1–3 orders of magnitude in inland areas that cannot be explained by rainfall variability alone.

The majority of $R_{50}$ values in our dataset are either low or moderate, between 1–10 mm y$^{-1}$ (35 %) or 10–100 mm y$^{-1}$ (38 %), respectively. Extremely low $R_{50}$ values (i.e., <1 mm y$^{-1}$) constitute 16 % of the dataset, while high $R_{50}$ values (i.e., 100–1,000

mm y$^{-1}$) constitute 11 % of the dataset. Only 0.01 % of $R_{50}$ values are extremely high (i.e., >1,000 mm y$^{-1}$). The point datasets of $R_{50}$, $R_5$ and $R_{95}$ before and after the data filtering process are available as electronic data files in the supporting information.

**3.3 Random Forest models and feature importance**

To explore the effects of the selection of variables in the random forest analyses (Table 1), different variable groupings were investigated as input features to train different $R_{50}$ random forest models. Table 2 outlines combinations of variables and their

impact on various fit metrics, showing the highest $R^2$ values, and lowest root mean square error (RMSE), mean absolute error (MAE), and the number of trees used.

**Table 2.** Best results from random forest $R_{50}$ models developed using different variable groupings, showing optimal number of trees in each forest, training score ($R^2$) external validation test score ($R^2$), root mean square error (RMSE), and mean absolute error (MAE), where P=precipitation, AI=aridity index, PET=potential evapotranspiration, KG=Köppen-Geiger, RS=rainfall seasonality, DTC=distance to coast,

RD=regolith depth, WTD=water table depth, SP=slope percentage, E=elevation, G=geology, SC=soil class, CP=clay percentage, SiP=silt



percentage, SaP=sand percentage, NDVI=normalised difference vegetation index, VC=vegetation category. * Denotes the model selected for further analyses.

| Model / groupings | No. of trees | Training score $R^2$ | Out-of-bag score $R^2$ | External validation | | |
|---|---|---|---|---|---|---|
| | | | | Test score $R^2$ | RMSE (mm y$^{-1}$) | MAE (mm y$^{-1}$) |
| All variables | 200 | 0.795 | 0.720 | 0.735 | 51.5 | 20.8 |
| **Categorical grouping** | | | | | | |
| Climate (P, AI, PET, KG, RS) | 150 | 0.718 | 0.688 | 0.705 | 54.4 | 22.9 |
| Surface/hydrogeological (DTC, RD, WTD, SP, E) | 350 | 0.549 | 0.424 | 0.429 | 75.7 | 35.6 |
| Geomorphological (G, SC, CP, SiP, SaP) | 250 | 0.508 | 0.470 | 0.476 | 72.5 | 35.8 |
| Vegetation (NDVI, VC) | 350 | 0.571 | 0.519 | 0.524 | 69.1 | 32.3 |
| **Highest performing 4–8 variable grouping** | | | | | | |
| P, RS, PET, E | 150 | 0.745 | 0.700 | 0.716 | 53.4 | 22.3 |
| P, RS, PET, E, DTC | 300 | 0.758 | 0.707 | 0.720 | 53.0 | 21.9 |
| P, RS, PET, E, DTC, NDVI | 250 | 0.756 | 0.708 | 0.724 | 52.6 | 21.8 |
| P, RS, PET, E, DTC, NDVI, CP | 200 | 0.775 | 0.715 | 0.731 | 52.0 | 21.1 |
| P, RS, PET, E, DTC, NDVI, CP, SC* | 250 | 0.772 | 0.716 | 0.732 | 51.9 | 21.1 |

The results in Table 2 have also been influenced by the selection of optimal hyperparameters, such as the number of trees,
maximum depth of trees, and maximum features. Aside from grouping variables categorically by climate, surface/hydrogeology, geomorphology, and vegetation, various other groupings ranging from four variables to eight variables were also explored. Exploring fewer input variables allows us to assess whether a model trained on less variables could achieve similar model accuracy while being less computationally expensive. The strongest performing 4–8 variable groups are shown in Table 2. The best performing 8-variable model trained with 250 trees achieves a training score $R^2$ of 0.772, an external
validation test score $R^2$ of 0.732, RMSE of 51.9 mm y$^{-1}$, and MAE of 21.1 mm y$^{-1}$ which are similar to the all-variable model (Table 2). Model accuracy does not improve when a ninth variable (either regolith depth, water table depth, geology, sand percentage, slope percentage, vegetation class, Köppen-Geiger, aridity index or silt percentage) was added (see Table S2 of the supporting information); hence, the best performing 8-variable model was chosen.





Table 2 demonstrates the importance of the climatological variables, for example, producing an external validation test score $R^2$ value of 0.705, similar to the maximum external validation test score obtained across all parameter combinations (0.735). The $R_{50}$ random forest model selected for further analyses, (the best performing 8-variable model) consists of the variables precipitation (P), rainfall seasonality (RS), potential evapotranspiration (PET), elevation (E), distance to coast (DTC),

normalised difference vegetation index (NDVI), clay percentage (CP), and soil class (SC) (bottom row, Table 2). This observation highlights that while the climatological variables are strong controls on recharge, other variables related to surface processes, hydrogeology, geomorphology and vegetation are also important. The vegetation model (containing variables NDVI and vegetation class) having the second highest score in the categorical groupings suggests that in Australia, vegetation could be a more important control on recharge compared to surface/hydrogeological and geomorphological variables.

Out of the 8 input variables used in our best performing $R_{50}$ random forest model, P, RS, PET, and NDVI are ranked highest as shown in the feature importance plot in Figure 4. The feature importance plots for the $R_5$ and $R_{95}$ random forest models are provided in Figure S4 and S5 of the supporting information, respectively. For comparison, the feature importance plot for the $R_{50}$ all-variable model is provided in Figure S6 of the supporting information.

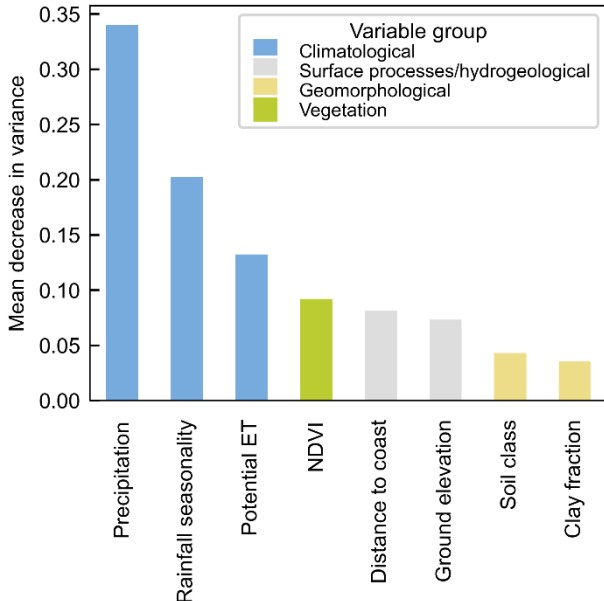

**Figure 4.** Mean feature importance through mean decrease in variance for the $R_{50}$ best performing 8-variable model (250 trees). The features are grouped according to climatological, surface processes/hydrogeological, geomorphological and vegetation variable groups depicted in Table 1.

The $R_{50}$ random forest model achieved a training score of $R^2$: 0.772, 'out-of-bag' score of $R^2$: 0.716, external validation test score of $R^2$: 0.732 and 10-fold cross validation $R^2$: 0.715, with 200 trees in the random forest (Figure 5). The relatively small

difference between the training score and external validation test score indicates that our model is not over-fitting the training



data. The similar $R^2$ values across different model evaluation methods indicate that our model should perform relatively well with unseen data. Figure 5a shows that our model tends to overestimate lower recharge values and underestimate higher values. Figure 5b further demonstrates this point. For example, for CMB recharge values between 0.001 mm y$^{-1}$ and 30 mm y$^{-1}$, our model tends to overestimate recharge, while at moderate to higher recharge rates (i.e., >30 mm y$^{-1}$) our model tends to

underestimate recharge. At high to extremely high recharge rates (i.e., >470 mm y$^{-1}$) our model only produces underestimates, which could be the result of underrepresentation of samples in extremely high recharge areas. The residuals at the higher end of recharge in Figure 5b may appear seemingly large, but the majority represent errors of less than 40 %.

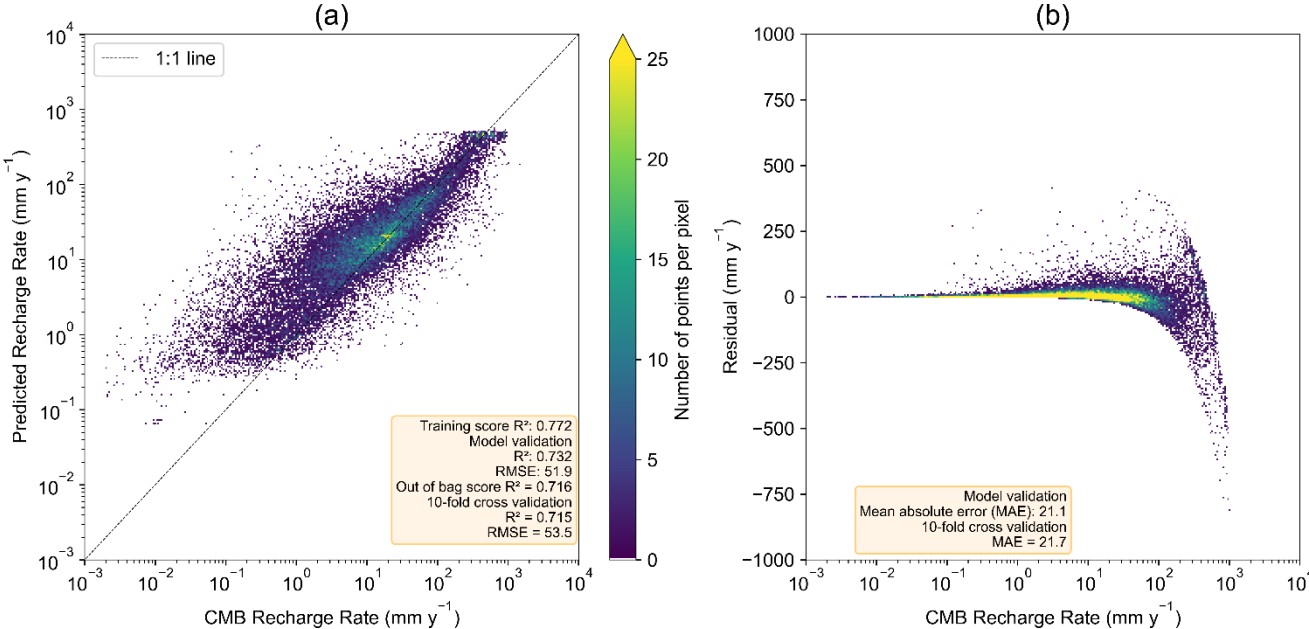

**Figure 5.** Model validation results for the selected $R_{50}$ model trained using 250 trees, showing: **(a)** CMB recharge rate ($R_{50}$) versus predicted
recharge rate, showing 1:1 line, and point density, and **(b)** CMB recharge rate ($R_{50}$) versus residuals (predicted recharge rate minus CMB recharge rate) and point density.

Compared to the µ$R_{50}$ of 43.5 mm y$^{-1}$ in Figure 2g, the RMSE of 51.9 mm y$^{-1}$ from external validation of our model (Figure 5a) might suggest relatively high variability and overall inaccuracy in model predictions. However, Figure 5a shows that most of the recharge rate estimates lie near the 1:1 line (as shown by the density of pixels in the colour map). When assessing only

$R_{50}$ <1 mm y$^{-1}$ for the validation results (Figure 5), we obtain an RMSE of 12.4 mm y$^{-1}$ or >1,000 %; however, percentage errors can be misleading when assessing errors of low values. This is similarly the case for $R_{50}$ from 1–10 mm y$^{-1}$ (RMSE: 19.4 mm y$^{-1}$), 10–100 mm y$^{-1}$ (RMSE: 29.8 mm y$^{-1}$), and 100–1,000 mm y$^{-1}$ (RMSE: 140.7 mm y$^{-1}$). Evaluating errors in different recharge ranges reveals that some errors are not as severe as they may appear. Model validation results for $R_5$ and $R_{95}$ recharge models are provided in Figure S7 of the supporting information.

The random forest generated groundwater recharge rate ($R_5$, $R_{50}$, $R_{95}$) maps of Australia (utilising P, RS, PET, E, DTC, NDVI, CP, and SC) are shown in Figure 6a, b and c.





**Figure 6.** Gridded groundwater recharge rate map of Australia generated using the highest performing random forest model, shown as **(a)** median recharge rate ($R_{50}$), **(b)** 95th percentile recharge rate ($R_{95}$) and **(c)** 5th percentile recharge rate ($R_5$) values, and gridded recharge ratio (R/P) map of Australia, shown as **(d)** $R_{50}$/P, **(e)** $R_{95}$/P and **(f)** $R_5$/P.

The CMB method provides recharge estimates that span the residence time of the groundwater (Crosbie et al., 2010a), hence the recharge outputs produced in Figure 6 represent recharge that has occurred over the longer term (e.g., hundreds to thousands of years). The variability in modelled recharge is highest within the arid Köppen-Geiger zones, which cover almost 80 % of the Australian continent, with $R_{50}$ ranging between ~0.03 and 278 mm $y^{-1}$, and a mean of 6.3 mm $y^{-1}$ (*n* pixels = 220,947). In the temperate Köppen-Geiger zones, which cover almost 12 % of the Australian continent, $R_{50}$ ranges between ~0.6 and 522



mm y$^{-1}$, with a mean of ~60 mm y$^{-1}$ ($n$ pixels = 33,177). In the tropical climates, which only cover 8 % of the Australian continent, R$_{50}$ ranges between and ~2.6 and 621 mm y$^{-1}$, with a mean of ~125 mm y$^{-1}$ ($n$ pixels = 22,897). As shown in Figure 6b and c, uncertainties in recharge estimates can range by orders of magnitude, regardless of climate zone. For example, the town of Tully, Queensland (located in the *Af* tropical Köppen-Geiger zone with latitude: -17.934°, longitude: 145.925°), has

the highest average rainfall in Australia (>3,100 mm y$^{-1}$) and the highest modelled R$_{50}$ of ~621 mm y$^{-1}$. However, the uncertainty ranges from 393 mm y$^{-1}$ to 1,759 mm y$^{-1}$. The town of Coober Pedy, South Australia (located in the *BWh* arid Köppen-Geiger zone with latitude: -29.012°, longitude: 134.753°), has one of the lowest average rainfalls in Australia (<150 mm y$^{-1}$), and a modelled R$_{50}$ of ~0.38 mm y$^{-1}$, with uncertainty ranging from 0.09 mm y$^{-1}$ to 0.56 mm y$^{-1}$.

The proportion of rainfall that becomes recharge, represented by the recharge ratios (R$_5$/P, R$_{50}$/P, and R$_{95}$/P) are shown as

gridded maps in Figure 6d, e and f, respectively. Like recharge, the variability in modelled R$_{50}$/P is the highest in the arid Köppen-Geiger zones, ranging over 4 orders of magnitude, from ~0.0001 to 0.42 (mean: 0.02, $n$ pixels = 220,947). In temperate and tropical climates, R$_{50}$/P ranges are smaller, from ~0.002 to 0.36 (mean: 0.06, $n$ pixels = 33,177) and ~0.003 to 0.35 (mean: 0.11, $n$ pixels = 22,897), respectively. The ranges in R/P reduce significantly when assessing the 5$^{th}$ and 95$^{th}$ percentiles (i.e., 90 % of the values are in the following ranges for arid, temperate and tropical zones: ~0.002–0.06, ~0.01–0.15, and ~0.03–

0.20, respectively). It should be noted that some values of R$_{95}$/P exceed a value of one due to the data filtering process only focused on removing bores with R/P >1 from the R$_{50}$ point recharge dataset. Therefore, both the R$_{95}$ gridded recharge and point recharge datasets will contain some unrepresentative recharge values with R/P values more than one. However, the number of values equates to <0.01 % of pixels in the R$_{95}$/P gridded map.

Boxplots showing the distribution of modelled recharge values (R$_{50}$, R$_5$ and R$_{95}$) and modelled recharge ratios (R$_5$/P, R$_{50}$/P,

R$_{95}$/P) categorised by arid, temperate and tropical Köppen-Geiger zones are shown as Figure S8 of the supporting information. The gridded maps of R$_{50}$, R$_5$ and R$_{95}$ are available as electronic text files in the supporting information.

# 4 Discussion

## 4.1 Groundwater recharge rate predictors

Clearly, precipitation has a strong control on groundwater recharge rates. While studies have found long-term average

precipitation to be the key predictor of recharge (e.g., MacDonald et al., 2021; West et al., 2023), others have found other precipitation-related factors such as aridity index (e.g., Berghuijs et al., 2022) or seasonal rainfall (e.g., Fu et al., 2019) to be the most important. Some investigations highlighted the strong explanatory power of vegetation and soils in addition to climate-related variables (e.g., Petheram et al., 2002; Crosbie et al., 2010a; Mohan et al., 2018; Moeck et al., 2020). Our R$_{50}$ random forest model incorporated eight variables from climatological, surface processes/hydrogeological, geomorphological

and vegetation categories. Using these eight variables in the feature importance analyses, our study revealed that the top four most important variables influencing recharge in Australia were precipitation (P), rainfall seasonality (RS), potential evapotranspiration (PET), and NDVI (Figure 4). These four variables highlight the importance of climatic factors on the





prediction of recharge, which agrees with other studies (e.g., Mohan et al., 2018; Berghuijs et al., 2022; West et al., 2023; Huang et al., 2023). Overall, the ranking of variables highlighted in our study is most aligned with the ranking of predictors in

Mohan et al. (2018), who found precipitation, PET and land use (vegetation) to be the top three important factors controlling recharge globally.

The aforementioned studies cover vastly different spatial scales, ranging from regional areas (e.g., Fu et al., 2019; Huang et al., 2023), the African continent (e.g., MacDonald et al., 2021; West et al., 2023), the Australian continent (e.g., Petheram et al., 2002; Crosbie et al., 2010a), to all continents (e.g., Mohan et al., 2018; Moeck et al., 2020; Berghuijs et al., 2022), and

contain datasets with varying spatial distributions and resolutions. The spatial variability across these previous studies suggests that some studies can have a climatic bias, depending on the climates included in the study area. For example, the chloride data used in our study to produce recharge estimates was mainly biased towards temperate and arid Köppen-Geiger zones (comprising ~50 % and ~40 % of the recharge dataset, respectively) and less towards tropical (~10 % of recharge values). The similarities and differences in climate types and recharge estimation techniques may influence the ultimate ranking of

important variables and be the reason for differences between studies.

It is important to highlight that while feature importance analyses can provide insight into important variables, overinterpretation should be avoided. Ranking of features in the feature importance plot can be affected by the choice of hyperparameters such as maximum features (e.g., limiting maximum features to a subset will avoid over-selection of the most important feature, such as precipitation in our case, during training of the random forest model). Feature importance may be

influenced by factors such as variable cardinality (i.e., tendency to score variables with many unique levels higher importance as they offer more opportunities for splitting the data; Strobl et al., 2007). Low cardinality of categorical features such as Köppen-Geiger, geology, soil class and vegetation class could be the reason for their relatively lower feature importance as shown in Figure S6 of the supporting information. Variables with lower importance can compete with more important variables, such that having more input variables does not necessarily improve performance of the model. Correlated variables

can also out-compete each other, leading to unreliable feature importance rankings (Toloşi and Lengauer, 2011). Some highly correlated variable pairs likely act as proxies for each other during the training process when the subset of features randomly selected only contains one of the variable pairs. Such is likely the reason for the climate group being most important in the all-variable model (Figure S6 of the supporting information). Similarly, the relationship between precipitation, distance to coast and elevation could explain why these variables also rank highly.

**4.2 Comparison of groundwater recharge rate estimates with previous studies**

The average groundwater recharge rate estimates produced for the Australian continent differs from those found in other studies, both for point recharge (Figure 3) and the modelled recharge (Figure 6). For example, the mean point recharge rate for the Australian studies collated by Crosbie et al. (2010a) was 257.2 mm y$^{-1}$ ($n$ = 4,360), compared to 43.5 mm y$^{-1}$ in our study ($n$ = 98,568). Similar mean recharge values of 246.5 mm y$^{-1}$ from Australian studies collated by Moeck et al. (2020; $n$ =

4,579) and 244 mm y$^{-1}$ from Berghuijs et al. (2022) were not surprising given that the data from Crosbie et al. (2010a) was



used in both studies. The mean recharge rate for the Australian studies collated by Mohan et al. (2018) was much closer to our study at 46.2 mm y$^{-1}$. This is likely due to the much smaller dataset of Mohan et al. (2018; $n = 217$) and limited spatial coverage – especially in tropical Northern Australia, compared to other studies.

The higher mean recharge values of the point data reported in other studies that cover Australia (e.g., Crosbie et al., 2010a; Moeck et al., 2020; Berghuijs et al., 2022) compared to ours can be attributed to the spatial distribution of recharge point estimates, and the estimation of recharge values using different recharge estimation techniques (i.e., water table fluctuation method, water balance, CMB, and other tracers). Studies that collated recharge estimates from other continents have also reported higher recharge rates than our point estimates. For example, MacDonald et al. (2021) reported median decadal point recharge estimates from compiled studies for different aridity zones in the African continent, with arid, semi-arid and humid

areas equivalent to 6 mm y$^{-1}$, 20 mm y$^{-1}$, and 130 mm y$^{-1}$, respectively. Point estimates of recharge from our study had median values of 1.1 mm y$^{-1}$, 8.0 mm y$^{-1}$, and 45.8 mm y$^{-1}$, for arid, semi-arid, and humid areas in Australia, respectively across these climate zones. This suggests that in the long term, aquifer systems in Australia are replenished on average at a rate 2–4 times lower than those in Africa.

Regarding the methods used, the CMB method produces long-term average diffuse groundwater recharge rates that are lower,

compared to other methods, including the water table fluctuation method, that estimate modern recharge. For example, methods such as the water table fluctuation method and tritium tend to estimate different recharge rates relative to those obtained via the CMB method, particularly in Australia, where modern recharge rates have increased due to large scale land clearing (Cartwright et al., 2007). Measurements using the water table fluctuation method will also be heavily influenced by focused recharge in areas where indirect recharge processes are dominant (e.g., leakage from ephemeral streams in arid regions;

Cuthbert et al., 2016) as opposed to diffuse recharge measured by the CMB method. These observations likely highlight the importance of considering recharge estimation type in the collation and use of large datasets. For example, recharge studies that have compared recharge estimation techniques have found large differences across different methods (e.g., Cartwright et al., 2007; King et al., 2017; Walker et al., 2019; Cartwright et al., 2020).

The mean modelled ($R_{50}$) recharge rate from our gridded recharge rate map was 22.7 mm y$^{-1}$, which is significantly lower than

modelled global estimates. For example, Mohan et al. (2018) reported a long-term, global average recharge of 134 mm y$^{-1}$, whereas Müller Schmied et al. (2021) reported a global mean diffuse recharge rate of 111 mm y$^{-1}$. The significant difference between these modelled recharge values is likely due to the large proportion of arid and semi-arid areas in Australia. Our gridded map contains 278,253 pixels of which ~80 % are in an arid Köppen-Geiger climate (see Figure S9 in the supporting information), compared to ~26 % of the global land area that is classified as arid (Gaur and Squires, 2018). The mean modelled

recharge for the Australian continent was not reported in either Mohan et al. (2018) or in Berghuijs et al. (2022). However, Berghuijs et al. (2022) highlight that their recharge estimates are higher than those presented in other global studies (e.g., Döll and Fiedler, 2008; de Graaf et al., 2015; Mohan et al., 2018; Müller Schmied et al., 2021), and are therefore, on average, likely to be higher than those presented here. We highlight that numerical outputs from these studies should be provided more





routinely. Sharing these numerical outputs could facilitate further comparisons and produce more useful outputs for potential
users.

## 4.3 Limitations and implications

In this study, the assumptions for estimating recharge using the CMB method were implemented through a data filtering process (Sect. 2.4), which was crucial to improving the reliability of inputs to our model. While we assume that erroneous recharge estimates have been removed during the data filtering process, some criteria that were assessed in other studies (e.g.,
Crosbie et al., 2022; Crosbie and Rachakonda, 2021) were not considered here due to the challenges of implementing them on a continental scale. For example, excluding measurements from bores screened within alluvium (e.g., Crosbie et al., 2022), would require a thorough understanding of local conceptual models and hydrogeological processes (e.g., cross-aquifer interaction) and existing recharge processes (e.g., flooding). By not excluding bores located in alluvium, point and modelled recharge estimates for these bores can be underestimated if additional chloride not sourced directly from rainfall is present.

The tendency of our model to underestimate recharge where moderate to higher recharge rates (i.e., 30–1,000 mm y$^{-1}$) were estimated from the CMB method, may be related to a skew in the distribution of our point recharge dataset towards lower recharge rates. The tendency for overestimation could be due to the aggregation of random forest leaf node values and tree predictions using the arithmetic mean which can be biased by large outlier values.

Large areas (e.g., inland Western Australia) had no chloride data and hence, the modelled recharge for these areas can be
subject to larger ranges of uncertainty. We do not account for geology in our model; therefore, modelled recharge rates can be significantly overestimated in areas such as where low permeability bedrock outcrops at the surface. Similarly, we highlight that users should be aware of the range of uncertainty in the modelled recharge when using values from the analyses presented here. The same message was emphasized by Leaney et al. (2011) and Crosbie et al. (2010a) for the 'method of last resort'. As is the case with all hydrogeological measurements and models, users of our modelled recharge rates should exercise expert
judgement and determine whether the estimates are reliable and fit-for-purpose. Preference should always be given to the collection of field data to constrain recharge estimates where possible.

For groundwater practitioners in Australia, our study provides an extensive database of groundwater chloride measurements and rigorously interpreted groundwater recharge rate estimates at high spatial resolution that holds potential for further use for researchers and water resource managers. We present a more robust, stochastic recharge rate estimator, modified from
CMBEAR (Irvine and Cartwright, 2022) to include the runoff coefficient term utilised in recent regional Australian studies (e.g., Crosbie et al., 2018; Crosbie and Rachakonda, 2021). Our study produced long-term recharge maps of the Australian continent. While Australian recharge maps have been produced previously (e.g., Leaney et al., 2011), this is the first time that a model of such scale has been developed on recharge estimates derived from only a single recharge estimation technique. Furthermore, by providing the Python code, point estimates and gridded map, we facilitate a transparent and reproducible
workflow that enables the broader community to utilise our methodology or further improve the approach.



## 5 Conclusions

We produce a groundwater recharge rate dataset for Australia with high resolution based on the chloride mass balance (CMB). This combines more than 200,000 compiled chloride measurements, existing chloride deposition maps, 17 national spatial gridded datasets, and a rigorous groundwater recharge rate estimation workflow. We enhance an open-source python tool,
CMBEAR and leverage existing methodologies (e.g., Crosbie et al., 2018) to provide an efficient, reproducible, and transparent stochastic approach that can be applied to anywhere in Australia. This approach quantifies uncertainty by creating recharge rate probability distributions, providing the $5^{th}$ and $95^{th}$ percentiles of point recharge rate estimates ($R_5$ and $R_{95}$) using distributions of groundwater chloride, runoff and chloride deposition.

We utilise subsets of the CMB recharge datasets ($R_5$, $R_{50}$ and $R_{95}$) to train and test three random forest regression models for
the purpose of upscaling point recharge estimates and assessing relative importance of recharge predictors. We show that climate-related variables (i.e., precipitation, rainfall seasonality and PET) have the strongest control on the groundwater recharge rate, but vegetation (NDVI) is also important. Other geographic and geomorphologic variables ranked lower but are still relatively important. The importance of climate and vegetation as recharge predictors are generally aligned with global recharge studies. The use of only eight of the 17 variables demonstrates that similar prediction performance can be achieved
with less variables, while reducing computation time and ensuring adequate performance on unseen data.

We present a gridded map of groundwater recharge rate estimates and uncertainties that could be valuable where data required to estimate recharge rates may be scarce or not available. Our groundwater recharge model utilises a data-driven approach based on a single recharge estimation technique to provide long-term recharge rates. Our CMB-based recharge rates are considerably lower than other studies including global water balance models (e.g., Döll and Fiedler, 2008; de Graaf et al.,
2015; Müller Schmied et al., 2021). This is likely due to the fact that global water balance models estimate modern recharge. We emphasise that the appropriate recharge timescales (e.g., long-term or modern) and mechanisms (e.g., diffuse or focused recharge) should be taken into consideration when collating recharge values produced from different techniques for the purpose of modelling recharge. We recommend that users exercise care and expert judgement when utilising the groundwater recharge rate estimates from these large-scale groundwater recharge models.

By applying the most widely used recharge estimation method (e.g., Moeck et al., 2020; Crosbie et al., 2010b), we provide a robust approach to automate the estimation of long-term diffuse recharge rates (with uncertainty). With chloride data being amongst the most common of groundwater analytes, there are significant opportunities to conduct similar analyses elsewhere.

*Code and data availability.* The code and output data presented in this paper is available as supporting information from
https://www.hydroshare.org/resource/088b1f35ee1b4c348a44a6cbad21250d/. Data presented in this paper has been visualised using scientific colour maps created by Crameri (2018). Gridded data inputs for the CMB recharge estimator Python code, including precipitation, chloride deposition, runoff coefficient, PET and aridity index are provided with attribution in the supporting information. Other gridded and non-gridded datasets used here can be downloaded from the references provided.



*Author contributions.* Conceptualisation: DI, CD, IC; software development: SL, DI; data preparation: SL, DI, GR; analyses: SL, DI, CD; writing – original draft: SL, DI; writing – review and editing: SL, DI, CD, GR, IC.

*Competing interests.* The authors declare that they have no conflict of interest.

*Acknowledgements.* We would like to acknowledge Geoscience Australia, CSIRO, the Bureau of Meteorology, and Visualising Victoria's Groundwater (Federation University) for making the data used in this study publicly available, and the institutions and individuals that collected the data originally. We thank Michelle Broad and Steven Tickell for their contribution of data. Stephen Lee was supported by a Research Training Program scholarship through Charles Darwin University. The research was supported by the CRC for Developing Northern Australia through the Water Security Program (AT.7.2223014).

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
