# Peer review of "A high-resolution map of diffuse groundwater recharge rates for Australia"

_EGUsphere, 2023_

## Author Comment (AC1)

**Response to Reviewer Comments – Reviewer 1**

We thank the editor and the three reviewers for their constructive comments and suggestions. We thank all three reviewers' comments that the text is well written and for their recognition of our study as a valuable resource for groundwater practitioners. We believe that in addressing their comments, the manuscript will be considerably improved and be ready for publication.

Most questions were about minor text updates and queries. Two reviewers asked for further comparisons between our outputs and previous investigations. We present a suggested approach to address these comments, including new figures for both the manuscript and the supporting information.

We believe that these additions directly address reviewer concerns, clearly showing the impact of the distribution of our estimates as a primary control on the differences in recharge estimates between our study and previous studies.

Our responses to the reviewer's comments (**RC**) are provided below as author's comments (**AC**). To help with the assessment of our responses, we colour coded our responses into agreement (green), partial agreement (yellow) and disagreement (red). When referring to text excerpts in our manuscript, we have provided the line number and whether , or if new text is added.

**RC1:** This paper presents an interested method to estimate groundwater recharge rates across Australia using chloride measurements. The text is well written and logically organized so that is easy to follow. The figures are excellent and ready for publication. I am not specialist of geochemistry or chloride but I found this study very relevant for hydrologist like me who is interested in groundwater processes. While I consider my following comments as minors, I think they must be addressed with attention before publication.

> **AC:** Thanks for your interest, positive feedback and helpful comments on our manuscript that intend to improve our work.

**RC2:** I am especially disappointed by the comparison between the presented product (from chloride) to the previous estimates from other studies (for example Moeck et al. 2020). Such previous studies report mean recharge estimates 5 times larger than the present study. You only attribute this drastic difference to "the spatial distribution of recharge point estimates, and the estimation of recharge values using different recharge estimation techniques" (line 470). I am sorry but I am not very convinced by this argument.

> **AC: We partially agree (minor changes to the manuscript and supporting information suggested).** To address this point, we will provide updates to the text (see below) and present a new figure in the manuscript and another in the supporting information to make this point more definitively. See suggested text changes, figure and supporting information figure below.
>
> **Suggested text and figure changes in manuscript**
>
> Suggested revision at line 469: The higher mean recharge values of the point data reported in other studies that cover Australia (e.g., Crosbie et al., 2010a; Moeck et al., 2020; Berghuijs et al., 2022) compared to ours can be attributed to the difference in spatial distribution of recharge point estimates, and the different recharge estimation methods used.

 Several differences in the method are important, including:

(1) 60 % of the estimates in Crosbie et al. (2010) and Moeck et al. (2020) were from an earlier study (Crosbie et al., 2009), which used a simpler CMB method and an older chloride deposition map to calculate recharge (see chloride deposition maps suggested for the supporting information, figure S9b).

(2) Our method incorporates the most recent improved chloride deposition map with enhanced data and spatial coverage (Wilkins et al., 2022).

(3) There are key differences in chloride deposition rates between the different chloride deposition maps, especially within 50 kilometres of the coastline, that can significantly affect the resulting recharge rate (see chloride deposition maps suggested for the supporting information, figure S9).

The mean of the 2,722 CMB recharge estimates from Crosbie et al. (2009) is 388 mm y$^{-1}$ while the mean of the other 1,620 estimates from Crosbie et al. (2010) which were estimated from 14 different methods (including 38 % from CMB, 25 % from transient soil CMB, and 9 % from water table fluctuation) is 40 mm y$^{-1}$. The estimates from Crosbie et al. (2009) are likely overestimates and were flagged by Crosbie et al. (2010) to have very little quality control.

(4) Our approach accounts for chloride lost to runoff in the calculation, resulting in a reduction in our recharge rates compared to the simpler method used in Crosbie et al. (2009) which does not consider this factor.

(5) Our methodology is stochastic, performing 1,000 recharge calculations to generate a probability distribution. We present the median and an error range taken as the 5$^{th}$ and 95$^{th}$ percentiles of the distribution to provide a more robust interpretation of the results.

The spatial distribution is important because the climate at the location of the recharge estimate strongly influences the annual recharge rate (Moeck et al., 2020). Figure 7 (new, below) demonstrates this point by using the climate zones found in Australia that are classified from different aridity index values (i.e., in order of increasing aridity or decreasing recharge potential: humid, dry subhumid, semi-arid, arid, and hyper-arid, based on United Nations Environment Programme, 1997).

Suggested new figure (Figure 7) is shown below:

[Figure]

**Figure 7 Maps and histograms showing the difference in spatial distribution and proportion (%) of the point recharge dataset of (a, d) Crosbie et al. (2010a), (b, e) Moeck et al. (2020) and (c, f) our study that are located in various aridity classes (Hyper-arid, arid, semi-arid, dry-subhumid and humid; United Nations Environment Programme, 1997). The proportion (%) and mean recharge (mm y⁻¹) are shown in the histograms above each bar.**

The proportion of recharge estimates from Crosbie et al. (2010a) and Moeck et al. (2020) located in dry subhumid and humid aridity classes is double that of our dataset (Figure 7). The mean recharge rates in Crosbie et al. (2010a) and Moeck et al. (2020) for each aridity category are all higher than our study – particularly dry subhumid and humid which are 3-4 times higher. The higher proportion of estimates in the dry subhumid and humid climate zones together with the significantly higher mean recharge rates in these climates, results in a higher overall mean recharge rate for the Crosbie et al. (2010a) and Moeck et al. (2020) datasets compared to our study.

Suggested revision at line 492: Our gridded map contains 278,253 pixels of which ~80 % are in an arid Köppen-Geiger climate (see Figure S11 in the supporting information), compared to ~26 % of the global land area that is classified as arid (Gaur and Squires, 2018).

**Suggested figure in supporting information**

Suggested new figure S9 (supporting information) shown below:

[Figure]

**Figure S9.** Maps showing gridded deposition maps from (a) Davies and Crosbie, (b) Crosbie et al. (2009), (c) Wilkins et al. (2022) and (d) Wilkins et al. (2022) and recharge sites that were able to be matched (green) and unable to be matched (red) to those in our study.

**RC3:** Why you do not compare point estimates from Moeck et al. (2020) with your product at the same locations, exactly? For example, using a squatter plot, or the density function of each product. In other words, I find your comparison with existing previous products is not enough detailed. Please, improve this comparison.

> **AC:** **We partially agree (minor changes to the manuscript and supporting information suggested).** Only a limited comparison can be made for several reasons, outlined below. This discussion will be added to the supporting information, with associated analyses.
>
> **Suggested text changes in manuscript**
>
> Suggested revision at line 472: Further details including limitations on the comparisons with Crosbie et al. (2010a) and Moeck et al. (2020) are provided in the supporting information. Studies that collated recharge estimates from other continents have also reported higher recharge rates than our point estimates.
>
> **Suggested text and figure changes in supporting information**
>
> Suggested new text and figure (S10) in the supporting information shown below:
>
> **6. Comparison of our point dataset with Moeck et al. (2020) and Crosbie et al. (2010a)**
>
> Only a limited comparison can be made between our point recharge dataset and Moeck et al. (2020) and Crosbie et al. (2010a) due to the following reasons:

1) The Moeck et al. (2020) dataset did not provide information on the estimation method or bore IDs, and only approximate location information. Thus, identifying specific bores to allow like-for-like analyses is not possible. Only 346 out of 4,579 Moeck et al. (2020) estimates could be approximately paired with a bore using matching latitudes and longitudes.

2) However, the Crosbie et al. (2010a) study (data contained within the Moeck et al. (2020) dataset) collated a dataset of Australian recharge estimates ($n$=4,360), presenting the data in a spreadsheet. This spreadsheet also included the recharge estimation method/technique used in the original study as well as in some cases, the bore ID.

3) Approximately 60 % of the Crosbie et al. (2010a) dataset was collated from Crosbie et al. (2009). These recharge estimates were flagged by Crosbie et al. (2010a) to have very little quality control. Upon review, we found that their methodology was too different for a meaningful comparison with our study (e.g., they utilise a simpler CMB method with a vastly different and recently revised chloride deposition map). I.e., we utilise recent chloride deposition maps from Wilkins et al. (2022).

For our comparison, we utilise the remaining data from Crosbie et al. (2010a), which comprise datasets from four recharge studies (i.e., Banks et al., 2007a, b; Green et al., 2007; Harrington et al., 1999). Figure S10 compares the recharge rates from these four studies to those matching in our study.

[Figure]

Figure S10. Comparison of recharge rates collated for Crosbie et al. (2010a) against median recharge rates ($R_{50}$) from our study. Grey open circles represent the mean recharge rates from Banks et al. (2007a) which plot closer to the 1:1 line than the minimum recharge rates collated in

**Crosbie et al. (2010a). Blue arrows show the deviation of Green et al. (2007) recharge rates from the 1:1 line due to the different chloride deposition values used in their study.**

The recharge estimates from (Harrington et al., 1999) and (Banks et al., 2007b) had similar recharge rates compared to our study (plotting close to the 1:1 line). However, our estimates were consistently higher compared to estimates from (Banks et al., 2007a). Our estimates were higher than those from the (Banks et al., 2007a) study, as Crosbie et al. (2010a) used the minimum estimates from that study (Fig. S10, solid grey circles. The mean recharge estimates (grey open circles, Fig. S10) effectively lie on our 1:1 line. Similarly, our estimates are consistently lower than those from Green et al. (2007). The large difference between our estimates and Green et al. (2007) can most likely be attributed to their methodology for calculating chloride deposition rates for their study. Further discussion below.

Green et al. (2007) used either an annual rainfall value of 800 mm y$^{-1}$ or 840 mm y$^{-1}$, along with a rainfall chloride concentration of 7.2 mg L$^{-1}$ or 8.4 mg L$^{-1}$, equating to a range of chloride deposition values between 57 kg ha$^{-1}$ y$^{-1}$ and 70.6 kg ha$^{-1}$ y$^{-1}$. The chloride deposition values used in Green et al. (2007) are approximately double those used in matching bores from our study, which range from 26.1 kg ha$^{-1}$ y$^{-1}$ to 38.0 kg ha$^{-1}$ y$^{-1}$. Their average rainfall chloride concentrations were calculated from 3-4 rainfall samples collected over a two-year period, making their chloride deposition rates less reliable. Our chloride deposition rates have been spatially extrapolated from rainfall gauges with a larger number of samples. Additionally, the runoff coefficients used in our study which ranged from 0.1 to 0.39 (average of 0.24) tended to be higher than the 0.1 blanket value used in Green et al. (2007). Both factors have most likely contributed to the recharge rates in Green et al. (2007) being double those calculated in our study.

**RC4:** It seems that, naturally, Australian soils are largely affected by salt (Wicke et al. 2011; https://www.encyclopedie-environnement.org/en/zoom/land-salinization/). Perhaps my understanding is not very good, but it seems that only chloride range of 35-125 mg/L for groundwater is considered as normal. Your Figure 1 shows that at least the half of your data are superior to this range. Could your underestimation (of recharge estimates) thus be due to a natural large groundwater concentration in chloride (Clgw) making your equation 2 obsolete?

> **AC: No change suggested.** We are unsure where the reviewer has obtained the chloride range of 35-125 mg/L from as it is not in the link provided. Groundwater in Australia commonly has high Cl concentrations largely due to moderate rainfall, the semi-arid climate, and high transpiration rates of the native vegetation (Allison et al., 1990; Cartwright et al., 2004), which results in most of the water being returned to the atmosphere and little recharge (which is what our analysis shows). Our analyses use ~100,000 measured values from a database managed by an Australian government organisation. We provide a detailed description of the process to omit unreliable data values.

**RC5:** Chloride concentration in groundwater could be significantly impacted by human activities like agriculture or industry. If anthropogenic chloride "flows" into groundwater, its concentration will be larger than in natural systems and then, because equation 2, your estimates will be biased and too low. Are you sure that your measurements are not drastically impacted by these processes?

**AC: We partially agree (minor changes to the manuscript suggested).** The reviewer correctly highlights that additional sources of chloride may cause our estimates to be lower. While salting of roads is not common practice in Australia, anthropogenic sources of chloride such as from irrigation water and fertilisers may alter groundwater chloride concentrations. Additionally, the majority of studies where chloride/bromide (Cl/Br) ratios are reported indicate that the Cl is from evaporation of rainfall and not from other sources, which would lead to elevated Cl/Br ratios.

**Suggested text changes in manuscript**

Suggested revision at line 508: By not excluding bores located in alluvium, point and modelled recharge estimates for these bores can be underestimated if additional chloride not sourced directly from rainfall is present, for example, through the application of irrigation water or chloride-based fertilisers (e.g., potassium chloride).

**RC6:** Line 550, you clam "Our CMB-based recharge rates are considerably lower than other studies including global water balance models (e.g., Döll and Fiedler, 2008; de Graaf et al., 2015; Müller Schmied et al., 2021). This is likely due to the fact that global water balance models estimate modern recharge." … Ah ok, but why, I do not understand your explanation? Do you mean that your estimates do not account for modern recharge, so they account for what? In other words, what is the period of validity of your estimates?

**AC: We partially agree (minor changes to the manuscript suggested).** We will address this comment in three ways: (1) We will include a new figure in the manuscript (Figure 7) to show that our dataset contains significantly more recharge estimates in the arid and semi-arid zones than other datasets, (2) we will include the text below relating to the timescales of CMB recharge estimates, and (3) the text below relating to land use change and the impacts on recharge estimation methods that operate on modern timescales.

**Suggested figure and text changes in manuscript**

(1) Suggested new figure (Figure 7) shown below:

[Figure]

**Figure 7** Histograms and maps showing the difference in spatial distribution and proportion (%) of the point recharge dataset of (a, d) Crosbie et al. (2010a), (b, e) Moeck et al. (2020) and (c, f) our study that are located in various aridity classes (Hyper-arid, arid, semi-arid, dry-subhumid and humid; United Nations Environment Programme, 1997). The proportion (%) and mean recharge (mm y⁻¹) are shown in the histograms above each bar.

(2) Suggested revision at line 550: This is likely due to the fact that CMB operates on longer timescales that span the residence time of the groundwater (e.g., chloride can take between 4,000 and 40,000 years to accumulate in the Murray Basin, South Australia).

(3) Suggested revision at line 550: Contrary to this, global water balance models estimate modern recharge (i.e., over the last century where climate and soil data are available). Recharge estimation methods operating over modern timescales tend to be more easily impacted by land-use change. For example, Scanlon et al. (2006) demonstrate groundwater recharge both pre-and post-clearing in an Australian context, showing significant change (increase) in recharge.

**RC7:** So, even if I found this study very relevant and promising, I think that (1) the comparison with other product should be done more in depth, (2) a discussion about the salt-affected soils over Australia is perhaps relevant, and (3) a discussion about the impact of anthropogenic processes on groundwater chloride concentration (and then on your results) must be emphasized.

> **AC: No (additional) change suggested.** We have addressed this comment in above responses.

**Additional references not already included in the manuscript**

Allison, G. B., Cook, P. G., Barnett, S. R., Walker, G. R., Jolly, I. D., and Hughes, M. W.: Land clearance and river salinisation in the western Murray Basin, Australia, Journal of Hydrology, 119, 1–20, https://doi.org/10.1016/0022-1694(90)90030-2, 1990.

Banks, E., Zulfic, D., and Love, A.: Groundwater Recharge Investigation in the Tookayerta Creek Catchment, South Australia, Government of South Australia Department of Water, Land and Biodiversity Conservation, 2007a.

Banks, E., Wilson, T., Green, G., and Love, A.: Groundwater Recharge Investigations in the Eastern Mount Lofty Ranges, South Australia, Government of South Australia Department of Water, Land and Biodiversity Conservation, 2007b.

Cartwright, I., Weaver, T. R., Fulton, S., Nichol, C., Reid, M., and Cheng, X.: Hydrogeochemical and isotopic constraints on the origins of dryland salinity, Murray Basin, Victoria, Australia, Applied Geochemistry, 19, 1233–1254, https://doi.org/10.1016/j.apgeochem.2003.12.006, 2004.

Crosbie, R., McCallum, J. L., and Harrington, G. A.: Estimation of groundwater recharge and discharge across northern Australia, in: 18th World IMACS Congress and MODSIM09 International Congress on Modelling and Simulation. Modelling and Simulation Society of Australia and New Zealand and International Association for Mathematics and Computers in Simulation, 3053–3059, 2009.

Green, G., Banks, E., Wilson, T., and Love, A.: Groundwater recharge and flow investigations in the Western Mount Lofty Ranges, South Australia, Government of South Australia Department of Water, Land and Biodiversity Conservation, 2007.

Harrington, G. A., Herczeg, A. L., and Cook, P. G.: Groundwater Sustainability and Water Quality in the Ti-Tree Basin, Central Australia, CSIRO, 1999.

United Nations Environment Programme: World Atlas of Desertification: Second Edition, 1997.

---

## Author Comment (AC2)

**Response to Reviewer Comments – Reviewer 2**

We thank the editor and the three reviewers for their constructive comments and suggestions. We thank all three reviewers' comments that the text is well written and for their recognition of our study as a valuable resource for groundwater practitioners. We believe that in addressing their comments, the manuscript will be considerably improved and be ready for publication.

Most questions were about minor text updates and queries. Two reviewers asked for further comparisons between our outputs and previous investigations. We present a suggested approach to address these comments, including new figures for both the manuscript and the supporting information.

We believe that these additions directly address reviewer concerns, clearly showing the impact of the distribution of our estimates as a primary control on the differences in recharge estimates between our study and previous studies.

Our responses to the reviewer's comments (**RC**) are provided below as author's comments (**AC**). To help with the assessment of our responses, we colour coded our responses into agreement (green), partial agreement (yellow) and disagreement (red). When referring to text excerpts in our manuscript, we have provided the line number and whether , or if new text is added.

**RC1:** I found the manuscript to be very interesting and, as a groundwater modelling practitioner, I expect it to be a valuable resource if published. I expect to use it as a source for initial model parameterisation of diffuse, rainfall derived-recharge fluxes and for providing a point of comparison and reference for groundwater models in Australia.

The document is well written and provides an excellent description of the methods used, the main findings and discusses interesting outcomes including the limitations in the approach.

> **AC:** Thanks for your interest, positive feedback and helpful comments on our manuscript that intend to improve our work.

**RC2:** I understand that point estimates of groundwater recharge have been obtained from chloride measured in groundwater bores by the Chloride Mass Balance method using gridded chloride deposition, runoff, and precipitation datasets. The point estimates have been integrated through a Random Forest analysis to produce a recharge model for the entire continent.

Although I have no experience or understanding of the Random Forest method, I assume that the R5, R50 and R95 distributions illustrated in Figure 6 illustrate the uncertainty associated of the Random Forest analysis and do not include the additional uncertainty of the Chloride Mass Balance estimates used to obtain the point estimates. In my opinion the text would be improved by a clarification of this point.

> **AC: We partially agree (minor change to manuscript suggested).** The reviewer's general understanding of how the point estimates of recharge have been obtained is correct. However, it appears that the reviewer has mistaken the $R_5$, $R_{50}$ and $R_{95}$ gridded maps shown in Figure 6 to represent the uncertainty associated with the Random Forest analysis. Rather, these are the outputs of the three separate random forest models and represent the uncertainty in the application of the CMB methodology (i.e., including uncertainty of groundwater chloride concentration, chloride deposition and runoff coefficient). To make this point clearer we suggest making the following addition.

**Suggested text changes in manuscript**

Suggested revision at line 183: A probability distribution was created for each bore by calculating recharge (R) 1,000 times using the 1,000 sampled replicates from the distributions of $Cl_{gw}$, D and α. To quantify the uncertainty in recharge estimates, the  median recharge ($R_{50}$), 95th percentile recharge ($R_{95}$) and 5th percentile recharge ($R_5$) values were calculated from each probability distribution and provided as outputs for each bore.

**RC3:** I found the comparison to similar published studies in Section 4.2 to be of particular interest. I was surprised at the apparent discrepancy between the average point recharge estimates from the current study and those collated from other recharge studies in Australia (specifically Crosbie et al. (2010a) and Moeck et al. (2020)). The current study provides average point recharge estimates that are about 5 times lower than those obtained from the other studies. The text suggests that different distributions of data used to derive the recharge estimates and the different methods used to calculate recharge (including watertable fluctuation, catchment scale water budgets and other environmental tracers) may be the factors that explain these discrepancies. Without further discussion and examples, I find it difficult to accept that these issues can explain the magnitude of the discrepancy. For example, I find it unlikely that the spatial distribution of data used for the current and previous studies will be significantly different. I assume they all rely on measurements made in groundwater bores, the total population of which being the same for all studies. The discussion also calls into question the reliability of the Chloride Mass Balance method when compared to other recharge estimation techniques.

> **AC: We partially agree (minor changes to the manuscript and supporting information suggested).** To address this point, we will provide updates to the text and present a new figure in the manuscript and another in the supporting information to make this point more definitively.
>
> The question from this reviewer is highly similar to a question from Reviewer 1. See response to RC2 from Reviewer 1, as the suggested changes directly address this question from Reviewer 2.

**RC4:** While not suggesting that significant revisions to the manuscript are necessary, I believe the paper would benefit from a more focussed, qualitative assessment of uncertainties included in the recharge distributions presented in Figure 6. This should not only address the uncertainty in the Random Forest model but also in the uncertainty associated with Chloride Mass Balance estimates themselves including the reliability of the datasets used to obtain the point estimates.

> **AC: No (additional) change suggested.** We have addressed this comment in the previous two responses above.

---

## Author Comment (AC3)

**Response to Reviewer Comments – Reviewer 3**

We thank the editor and the three reviewers for their constructive comments and suggestions. We thank all three reviewers' comments that the text is well written and for their recognition of our study as a valuable resource for groundwater practitioners. We believe that in addressing their comments, the manuscript will be considerably improved and be ready for publication.

Most questions were about minor text updates and queries. Two reviewers asked for further comparisons between our outputs and previous investigations. We present a suggested approach to address these comments, including new figures for both the manuscript and the supporting information.

We believe that these additions directly address reviewer concerns, clearly showing the impact of the distribution of our estimates as a primary control on the differences in recharge estimates between our study and previous studies.

Our responses to the reviewer's comments (**RC**) are provided below as author's comments (**AC**). To help with the assessment of our responses, we colour coded our responses into agreement (green), partial agreement (yellow) and disagreement (red). When referring to text excerpts in our manuscript, we have provided the line number and whether , or if new text is added.

**RC1:** In this study, the authors used chloride mass balance (CMB) to derive long-term groundwater recharge rate estimates for Australia. A random forest model was built and tested using 17 relevant climatological, geologic, hydrologic, and static soil/vegetation variables as the predictors. The random forest model was validated using the point-scale CMB recharge rate estimates and the best-performing model was based on 8 of the 17 variables. The 8-variable model was used to generate the median, 5th, and 95th percentiles of recharge rate for the entire Australia. Overall, the manuscript is very well written. The experiments are set up in an organized and thoughtful way. I enjoy reading the discussion section where the authors provide guidance for practitioners to use the dataset. I only have some major and minor comments as outlined below.

> **AC:** Thanks for your interest, positive feedback, attention to detail and helpful comments on our manuscript that intend to improve our work.

**RC2:** Major comment:

Table 1: I have questions on the temporal evolution of these factors and the importance of the temporal component of the model. Depth to water table is a time-varying variable. Specify what value of depth to water table is used in this study.

> **AC: We agree (minor change to the manuscript suggested).**
>
> **Suggested addition to Table 1 of manuscript**
>
> Suggested addition to the "Description" column of the "Depth to water table" row in Table 1: Output of global numerical groundwater model. Mean simulated water table depth.

**RC3:** CMB is a method that measures long term (hundreds to thousands of years) groundwater recharge rate. I notice the authors use different time periods for different input features. My two questions are 1) Are those periods the longest time periods with data availability? 2) If yes to question 1), the time periods of data availability are still time periods that cannot match up the residence time of chloride which is on the order of hundreds to

thousands of years). How did the authors go about that? The authors can do a sensitivity analysis on using different time periods of input variable to test the sensitivity of their model results to the choice of input time periods.

> **AC: We partially agree (no change suggested).** Yes, we have chosen the input datasets with the longest time periods available. The reviewer correctly highlights the residence time of chloride and that the data available and used in our study still would not perfectly match up the residence time of chloride. However, for the large scale of our study, the datasets used (i.e., rainfall, runoff, evapotranspiration, etc.) are the best available datasets to use for the CMB method. We believe that a sensitivity analysis on using different time periods would not be required for this study as: (1) The residence time of chloride is on the order of hundreds to thousands of years, (2) we utilise the datasets with the longest time period available, (3) testing the sensitivity of shorter time periods (i.e., decadal time period) would not be appropriate for the long residence time of chloride (i.e., minimum of hundreds of years).

**RC4:** Minor comments:

Line 105: "…we identified 17 different gridded datasets (Table 1)." Is the distance to coast a gridded dataset? If it is a gridded dataset, specify the spatial resolution in Table 1. Otherwise, change the wording on Line 105.

> **AC: We agree (minor changes to the manuscript suggested).**
>
> **Suggested text changes in manuscript**
>
> Suggested revision at line 105: To investigate factors that influence groundwater recharge, we identified 17 different spatial datasets – 16 of which are available as gridded maps  (Table 1).

**RC5:** Table 1: The categories do not make total sense to me. Geology seems to belong to "Surface processes and hydrogeological" category. "geomorphological" can be changed to "soil properties". For sand, silt, and clay fractions, the description states they are 100 to 200 cm interval. Does this mean the input features are for the 100 -200 cm vertical layer? If yes, justify why choosing a deeper soil layer instead of values for the entire soil column.

> **AC: We agree (minor changes to the manuscript suggested) with the changes to Table 1 and Table 2. We partially agree with the latter part of the comment (no change made).**
>
> **Suggested changes to table in manuscript**
>
> Suggested revision in Table 1: Move Geology under the "Surface processes and hydrogeological" category. Change "geomorphological" category to "soil properties".
>
> Suggested revision in Table 2: Move Geology under the "Surface processes and hydrogeological" category. Change "geomorphological" category to "soil properties". Change all mentions of "geomorphological" or "geomorphology" to "soil properties".
>
> **Author comments**
>
> Yes, the input datasets for sand, silt and clay fractions are for the 100 to 200 cm vertical layer. This layer was chosen as it was the largest interval available out of a range of different intervals (i.e., 0 to 5 cm, 5 to 15 cm, 15 to 30 cm, 30 to 60 cm, 60 to 100 cm, and 100 to 200 cm), and therefore, would be the most effective interval at controlling recharge.

**RC6:** Line 205: Step (6) removes cases where estimated recharge equals or exceeds mean annual rainfall. Explain why and how did that happen. Could this be related to the errors underlying the estimation of recharge rates?

> **AC: We partially agree (no change suggested).** Step 6 in our data filtering process was implemented to remove recharge estimates that are unusually higher than the mean annual rainfall available at the location. There are three plausible reasons how such recharge estimates could occur: (1) under-estimation of groundwater chloride concentration used in our study due to limited measurements not being representative of actual long-term average, (2) error in the chloride deposition map used in our study (i.e., over-estimate of chloride deposition at recharge site), (3) the recharge estimate is in an area that receives additional sources of water that has significantly lowered the concentration of groundwater chloride (e.g., in an irrigation area). More detailed information was provided in the supporting information and referred to in line 206 of the manuscript.

**RC7:** Line 220: What is the "typical practice"? Specify the name of the method.

> **AC: We agree (minor change to the manuscript suggested).**

> **Suggested text changes to manuscript**

> Suggested revision at line 219: The dataset was split into a randomly selected training subset (70 %) and validation subset (remaining 30 %) following the train test split procedure  (e.g., West et al., (2023); Sihag et al., 2020; Rahmati et al., 2016).

**RC8:** Line 221: Each tree in the random forest model (the model) was trained on n randomly selected observations, with replacement (i.e., bootstrapping) from the training subset, where n is equal to the total number of observations in the training subset.

> **AC: We disagree (no change suggested).** We are unsure what the reviewer was implying with this comment as the reviewer has presented text verbatim from the manuscript. We note that we apply the random forest analyses in a routine way.

**RC9:** Line 306: "The recharge area of these deep systems is likely to be hundreds of kilometres away from the bore location, whereas our analyses assume recharge occurs within the 0.05° × 0.05° pixel from the chloride deposition map that contains the bore." How does this influence the results or how do the authors address this question.

> **AC: We disagree (no change suggested).** We believe this has been addressed in line 304 to line 306. As our study aims to estimate recharge for the shallow water table aquifer system which we assume is receiving recharge at or close to the groundwater bore location, we filter out groundwater chloride samples with sample depths more than 150 metres below ground surface, following previous published examples of this threshold. The filtered out deep samples are the ones that we believe recharge likely occurs "hundreds of kilometres away from the bore location…", and hence are omitted from the analyses.

**RC10:** Line 314: "The mean recharge rate…", do the authors mean "spatial mean recharge rate"?

> **AC: We partially agree (minor changes to the manuscript suggested).** We have addressed this with a suggested revision.

> **Suggested text changes in manuscript**

Suggested revision at line 314: The mean values of recharge rates for $R_{50}$, $R_{95}$, and $R_5$ (i.e., the point datasets) are 43.5 mm y$^{-1}$, 113.4 mm y$^{-1}$, and 25.8 mm y$^{-1}$, respectively.

**RC11:** Figure 3 and Line 320: Add the map of precipitation to Figure 3 to assist the comparison.

**AC:** We partially agree (minor changes to the manuscript suggested). We believe referring to a map of precipitation would assist in the comparison; however, rather than adding it to Figure 3, we suggest the following revision to the text.

**Suggested text changes in manuscript**

Suggested revision at line 319: As expected, high recharge rates are mostly located in areas with high precipitation, i.e., in the tropical north, along the east coast, and in north-western Tasmania (see Figure 3 and rainfall map in Figure S1a of the supporting information), while low recharge rates are mostly located inland from the coast.

**RC12:** Table 2 and Line 344: The best-performing 7-variable grouping has a performance as good as the 8-variable grouping. The less the number of variables, the lower the computation cost and potential of over-fitting. Why not choose the 7-variable grouping?

**AC:** We partially agree (no change suggested). The reviewer is correct in highlighting that the less the number of variables, the lower the computation cost; however, this is not true for the potential of over-fitting. We favoured the model not over-fitting the training data and the model's performance on unseen data rather than lower computational cost (which is not an issue for 8 variables at 200 to 250 trees).

**RC13:** Line 369: typo, "200 trees" should be "250 trees"

**AC:** We agree (minor changes to the manuscript suggested). We thank the reviewer for their attention to detail.

**Suggested text changes in manuscript**

Suggested revision at line 368: The $R_{50}$ random forest model achieved a training score of $R^2$: 0.772, 'out-of-bag' score of $R^2$: 0.716, external validation test score of $R^2$: 0.732 and 10-fold cross validation $R^2$: 0.715, with 200 250 trees in the random forest (Figure 5).

**RC14:** Line 445-459: Could the covariance/correlation between variables influence the feature importance of a specific variable? For example, precipitation, distance to coast, and elevation are correlated. Will using all three variables in the model introduce redundant information and potentially increase their explanatory power?

**AC:** We partially agree (no changes suggested). The reviewer is correct in highlighting that the covariance/correlation between variables could influence the feature importance of a specific variable. To use the reviewer's example, if precipitation is the strongest variable out of the group of correlated variables, then precipitation is likely to be chosen more often in a scenario where all three correlated variables are randomly chosen at a split in the random forest tree. If this was the case, then it would affect the ranking of these variables, i.e., precipitation may rank higher than the others. However, the random selection of variables at each split in the random forest tree (which is part of the random forest algorithm/method) is in place to limit these kinds of biases. Using all three variables may increase their explanatory

power (i.e., higher ranking in the feature importance); however, it does not introduce redundant information. Rather, it produces a better performing model as the algorithm always chooses the variable that splits the data the best.

**RC15:** Line 450: The sentence in the parenthesis reads weird. Rephrase.

> **AC:** **We agree (minor changes to the manuscript suggested).**
>
> **Suggested text changes in manuscript**
>
> Suggested revision at line 449: Feature importance may be influenced by factors such as variable cardinality (i.e., tendency to  give higher importance to variables with many unique levels  as they offer more opportunities for splitting the data; Strobl et al., 2007).

**RC16:** Line 515: Geology seems to be an important factor to explain the overestimation in the model. However, on line 450, the authors state that geology was not included in the highest-performing model because it cannot split the data due to low cardinality. It reads conflict to me. The authors should explain more.

> **AC:** We p**artially agree (minor changes to the manuscript suggested).** In line 515 we are suggesting that geology may be important to limit recharge where low permeability bedrock outcrops at or sub-crops close to the ground surface. However, the geology spatial dataset does not provide sufficient detail to differentiate between low permeability bedrock and more permeable, highly fractured bedrock. Therefore, we believe a revision to the text will remove this misunderstanding.
>
> **Suggested text changes in manuscript**
>
> Suggested revision at line 515:  No geological dataset is available that provides detailed spatial information on the permeability of bedrock; therefore, modelled recharge rates can be significantly overestimated in areas such as where low permeability bedrock outcrops at the surface and underestimated in areas where highly fractured bedrock exists.